# Mitogenomes Reveal Two Major Influxes of Papuan Ancestry across Wallacea Following the Last Glacial Maximum and Austronesian Contact

**DOI:** 10.3390/genes12070965

**Published:** 2021-06-24

**Authors:** Gludhug A. Purnomo, Kieren J. Mitchell, Sue O’Connor, Shimona Kealy, Leonard Taufik, Sophie Schiller, Adam Rohrlach, Alan Cooper, Bastien Llamas, Herawati Sudoyo, João C. Teixeira, Raymond Tobler

**Affiliations:** 1Australian Centre for Ancient DNA, School of Biological Sciences, University of Adelaide, Adelaide 5005, Australia; kieren.mitchell@adelaide.edu.au (K.J.M.); bastien.llamas@adelaide.edu.au (B.L.); 2Genome Diversity and Diseases Laboratory, Eijkman Institute of Molecular Biology, Jakarta 10430, Indonesia; leonard@eijkman.go.id (L.T.); herawati@eijkman.go.id (H.S.); 3ARC Centre of Excellence for Australian Biodiversity and Heritage, University of Adelaide, Adelaide 5005, Australia; 4Archaeology and Natural History, School of Culture, History and Language, College of Asia and the Pacific, Australian National University, Canberra 2601, Australia; sue.oconnor@anu.edu.au (S.O.); shimona.kealy@anu.edu.au (S.K.); 5ARC Centre of Excellence for Australian Biodiversity and Heritage, Australian National University, Canberra 2601, Australia; 6Evolution of Cultural Diversity Initiative, Australian National University, Canberra 2601, Australia; 7ARC Centre of Excellence for Mathematical and Statistical Frontiers, School of Mathematical Sciences, University of Adelaide, Adelaide 5005, Australia; sophie.schiller@adelaide.edu.au (S.S.); rohrlach@shh.mpg.de (A.R.); 8Department of Archaeogenetics, Max Planck Institute for the Science of Human History, 07745 Jena, Germany; 9Blue Sky Genetics, P.O. Box 287, Adelaide 5137, Australia; alanjcooper42@gmail.com; 10South Australian Museum, Adelaide 5000, Australia; 11Environment Institute, University of Adelaide, Adelaide 5005, Australia; 12National Centre for Indigenous Genomics, Australian National University, Canberra 2601, Australia; 13Department of Medical Biology, Faculty of Medicine, University of Indonesia, Jakarta 10430, Indonesia; 14Sydney Medical School, University of Sydney, Sydney 2050, Australia

**Keywords:** phylogeography, human migrations, Sahul, mitochondria, mtDNA

## Abstract

The tropical archipelago of Wallacea contains thousands of individual islands interspersed between mainland Asia and Near Oceania, and marks the location of a series of ancient oceanic voyages leading to the peopling of Sahul—i.e., the former continent that joined Australia and New Guinea at a time of lowered sea level—by 50,000 years ago. Despite the apparent deep antiquity of human presence in Wallacea, prior population history research in this region has been hampered by patchy archaeological and genetic records and is largely concentrated upon more recent history that follows the arrival of Austronesian seafarers ~3000–4000 years ago (3–4 ka). To shed light on the deeper history of Wallacea and its connections with New Guinea and Australia, we performed phylogeographic analyses on 656 whole mitogenomes from these three regions, including 186 new samples from eight Wallacean islands and three West Papuan populations. Our results point to a surprisingly dynamic population history in Wallacea, marked by two periods of extensive demographic change concentrated around the Last Glacial Maximum ~15 ka and post-Austronesian contact ~3 ka. These changes appear to have greatly diminished genetic signals informative about the original peopling of Sahul, and have important implications for our current understanding of the population history of the region.

## 1. Introduction

The rapid growth of human population genetic research over the past two decades has greatly improved our understanding of genetic structure and diversity of contemporary human populations and provided novel insights into how and when our ancestors colonised the planet. Cumulative evidence from modern and ancient anatomically modern human (AMH) genomes strongly suggests that all contemporary non-African human populations descend from a group of AMH migrants that left Africa around 50–60 thousand years ago (ka) before separating into multiple subpopulations that went on to populate all terrestrial regions of the planet, other than Antarctica [1]. These historical migrations have largely shaped the continental-level genetic structuring now apparent in present-day human genomes [2,3]. 

Genetic evidence also implies that one branch of these ancient migrations brought AMHs to Sahul by ~50 ka ago [4,5,6,7]. The Indigenous peoples of Australia and New Guinea share a deep genetic ancestry that stretches back to the initial peopling of Sahul [5,6,7], and they have since diversified into two genetically distinct groups suggestive of a separation that minimally dates back tens of thousands of years [4,5,6]. Intriguingly, while the majority of deep archaeological dates in Sahul align with the initial AMH occupation at around 50 ka ago [8], the Madjedbebe site in northern Australia points to AMH arriving in Sahul perhaps as early as 65 ka ago [9,10]. 

Regardless of the exact time of entry of AMH into Sahul, their movement from mainland Asia into Sahul required maritime traversal of the Wallacean archipelago—a series of more than 1000 individual islands distributed between the continental shelves of Sunda and Sahul—which now comprise the eastern section of modern Indonesia (Figure 1). Wallacea is interspersed by multiple fast moving ocean passages that have served as a natural barrier to faunal dispersal throughout the Pleistocene, resulting in the emergence of two distinct biogeographic zones that now distinguish the biota of mainland Asia from that of New Guinea and Australia. The initial maritime traversal of Wallacea by AMH constitutes the oldest recognised successful open-water crossings for *Homo sapiens* [11], although this region was also occupied by other hominin groups from the mid-Pleistocene (~700 ka [12]). The island paths taken by AMH en route to Sahul remain a key area of academic debate, with recent paleo-ecological and inter-island visibility modelling studies producing mixed support for either a northern route through Sulawesi and the Moluccas [13,14] or a southern route along the Lesser Sunda island chain [15] (Figure 1), whereby both routes remain plausible candidates [16]. 

Efforts to reconstruct the possible AMH migratory corridor(s) through Wallacea are complicated by the relatively sparse archaeological record of this region, which is further exacerbated by the submersion of land following the Last Glacial Maximum (LGM) that was likely exposed during the proposed period of AMH arrival in Wallacea > 50 ka ago [17]. Indeed, the oldest accepted AMH records in the region—between ~45 and 47 ka ago [18,19,20,21]—are more recent than the deepest dates reported for Australia and New Guinea, and, therefore, are not directly informative about this process. The patchy Wallacean archaeological record is complemented by a similar lack of regional genetic datasets; despite the growth of genetic sampling across Indonesia, many presently inhabited Wallacean islands are either unsampled, or available genetic data are limited to the hypervariable mitochondrial regions [22,23,24]. Moreover, recent genomic studies have largely focused on the more recent population history of the region, most notably the substantial genetic impact of Austronesian-speaking seafarers that mixed with the ancestors of modern Indonesian populations ~3–4 ka ago [25,26,27]. The general absence of detailed genetic and archaeological records for much of Wallacea means that little is currently known about the early human population history of this region and its role as a migratory corridor during the peopling of Sahul.

To learn more about the AMH population history of Wallacea and Sahul, we combined 351 newly sequenced whole mitogenomes from 11 Wallacean and West Papuan populations (Figure 1) with >400 mitogenomes from New Guinea and Australia and performed comprehensive phylogeographic analyses for the combined Wallacea–Sahul region. Our analyses provide compelling evidence for a surprisingly dynamic population history in Wallacea, which has important implications for our understanding of the long-term AMH inhabitation of this region.

## 2. Materials and Methods

### 2.1. Sample Collection and Ethics

Permission to conduct the research was granted by the National Agency for Research and Innovation, under the auspices of the Indonesian State Ministry of Research and Technology. Informed consent was obtained for the collection and use of all biological samples during community visits that were overseen by the Eijkman Institute for Molecular Biology team, following the Protection of Human Subjects protocol established by Eijkman Institute Research Ethics Commission (EIREC). All consenting participants were surveyed for their current residence, familial birthplaces, date and place of birth, and genealogical information for the three to four preceding generations. In total, informed consent was obtained for 351 individuals from 11 different populations across Wallacea (i.e., Kei, *n* = 30; Aru, *n* = 27; Tanimbar, *n* = 34; Seram, *n* = 34; Ternate, *n* = 30; Sanana, *n* = 25; Vayu (in Central Sulawesi), *n* = 23; and Rote-Ndao, *n* = 29) and West Papua (Keerom, *n* = 62; Mappi, *n* = 28; and Sorong, *n* = 30) (Figure 1 and Appendix A). Notably, the mitogenomes from the eight Wallacean populations and two of the three West Papuan populations (i.e., Keerom and Sorong) represent the first genetic information to be published for these groups.

### 2.2. Mitochondrial Sequence Generation

Genomes were extracted from the whole-blood samples using the Gentra Puregene Blood Core Kit C (QIAGEN, Hilden, Germany) following the manufacturer’s protocol. For each sample, two 8–9 kb mitochondrial amplicons were amplified, using the following two primer sets (sequences shown in 5′ to 3′ orientation): fragment 1, forward primer = CCCTATTAACCACTCACGGGAGC, reverse primer = CCAATTAGGTGCATGAGTAGGTGG; fragment 2, forward primer = ATCTGTTCGCTTCATTCATTGCCC, reverse primer = ACGCCGGCTTCTATTGACTTGGG. Each 25 μL PCR contained 1× KOD-Plus-Neo (TOYOBO, Osaka, Japan) buffer, 1.5 mM MgSO_4_, 0.2 mM dNTPS, 0.32 mM of each primer fragment 1 or 0.16 mM of each primer fragment 2, 0.5 U of KOD-Plus-Neo Enzyme, and 2 μL of DNA template. The cycling conditions were 94 °C for 2 min for initial denaturation, followed by 30 cycles at 98 °C for 10 s, then 69 °C for 5 min, for annealing and amplification, and a final 10 min extension step at 69 °C. Amplicons were checked using an agarose gel, purified using PCR DNA Fragments Extraction Kit (Geneaid, New Taipei, Taiwan), and pooled in equimolar concentration after being quantified via a Qubit dsDNA BR Assay Kit (Thermo Fisher Scientific, Waltham, MA, USA). For each sample, dual index mtDNA libraries were prepared using Nextera XT DNA Library Prep Kit (Illumina, San Diego, CA, USA), and library concentrations were quantified using Qubit dsDNA BR Assay Kit (Thermo Fisher Scientific, Waltham, MA, USA). Libraries were multiplexed into three separate pools, and 150 bp paired-end sequencing was performed according to the standard protocol for Miseq Reagent Kit V2 (2 × 151 bp).

Sequenced reads were demultiplexed using bcl2fastq2 v2.20 (Illumina, San Diego, CA, USA), with adapter removal and paired-end merging performed using fastp [28]. To minimise erroneous mapping of reads from nuclear mitochondrial sequences (Numt’s), the resulting merged reads were aligned to the human reference genome sequence (hs37d5) using BWA mem v0.7.15 [29]. Post-alignment processing involved indel realignment using GATK v3.5 IndelRealigner (https://hub.docker.com/layers/broadinstitute/gatk3/3.5-0, accessed on 24 April 2017), PCR duplicate removal via Markduplicates from Picard v2.2 (http://broadinstitute.github.io/picard, accessed on 24 April 2019) and base recalibration was performed using GATK BaseRecalibrator [30]. For each sample, we obtained between 30× and 400× mean coverage across the whole mitogenome sequence. Variant calling was performed using Freebayes v1.0.2 [31], following the removal of reads with mapping quality ≤25 and sites with base quality ≤20. BCFtools [32] was used to generate consensus sequences for each sample using the revised Cambridge Reference Sequence (rCRS [33]) to determine the reference alleles. Lastly, mitochondrial haplogroups were called from the consensus sequence using Haplogrep 2 [34], with sequence information from Phylotree build 17 [35]. Samples with identical haplotypes were limited to a single sequence, resulting in a total of 342 unique haplotypes being determined from the 351 original samples. 

### 2.3. Combined Wallacea–Sahul Dataset

Because we are primarily interested in investigating historical migrations between Wallacea and Sahul, we merged our mtDNA samples with previously published mitochondrial sequences sourced from other Indigenous populations from Indonesia, Australia, New Guinea, and Island Melanesia, after removing all mitogenomes affiliated with either Austronesian or mainland Asiatic ancestry from our data. Because the genealogical information contained in Austronesian/Asiatic haplogroups largely arose prior to the arrival of these lineages to Wallacea or New Guinea, it captures genealogical information that is not relevant for the current study. The final combined dataset contained 656 mitogenomes and included 186 of the 351 (~53%) new samples from the West Papua and Wallacea region (see Section 3 and Appendix A). We used MAAFT v7.407 [36] to create a multiple alignment for all 656 haplotypes to the rCRS reference. Indel removal was subsequently performed using Trimal v1.2 [37]. To reduce erroneous inferences, mitogenome regions identified as problematic on the Phylotree website were excluded from phylogenetic tree reconstruction (i.e., point mutations at positions 16,182, 16,183, and 16,519; insertions at 309.1, 315.1, and 16,193.1; and AC indels between 515 and 522 [35]).

### 2.4. Phylogenetic Parameter Estimation

To infer the genealogical relationships for the complete set of 656 mitogenomes, we used BEAST v2.6.3 [38] implementing an extended Bayesian skyline population history (to account for any temporal fluctuations in the effective population size) and a relaxed uncorrelated log normal clock (to allow rate heterogeneity between lineages). The bModel test option [39] was used to infer site model parameters (i.e., estimation of site-specific substitution rates, rate heterogeneity, and invariable site proportion). Monophyletic constraints were set for several major haplogroups (P, Q, R14, S, O, M73, M42, and N13), and the ancient sample KU659023 was assigned a tip date of 1250 years before present (yBP) according to the radiocarbon point estimate [40]. BEAST analyses were run using the nucleotide mutation rate estimate, μ, from Posth et al. [41], i.e., μ = 2.74 × 10^−8^ substitutions per site per year with s.d. = 2 × 10^−8^. We ran a total of 110 million Markov chain Monte Carlo (MCMC) iterations, with a burn-in period consisting of 10 million iterations and sampling performed every 10,000 steps thereafter. This resulted in 10,000 posterior samples that were used for further analyses. Manual inspection of parameters was performed using Tracer v1.7.1 [42], and all parameters exhibited appropriate Markov chain convergence and sufficient levels of sampling (i.e., effective sample sizes > 200 after discarding the 10% burn-in). Lastly, a maximum clade credibility (MCC) consensus tree was generated in TreeAnnotator v2.6.3 using the Common Ancestor Heights rule [43].

### 2.5. Using Ancestral Node Dates from Geographically Exclusive Clades to Infer Demographic History

To quantify the timing and direction of movements of mitochondrial lineages across Wallacea, Australia, and New Guinea, we extracted the time to the most recent common ancestor (TMRCA) for all well-supported geographically exclusive clades (GECs) within the posterior BEAST trees. We assumed that GECs represent regionally endemic matrilines and their TMRCAs provide estimates of the minimum bounds on the occupancy of the region. Because migration to a particular region must predate the TMRCA of GECs, the clustering of TMRCAs for particular regions provides evidence for a concentrated migration event around that time. We assume that our sampling was comprehensive enough that our mitogenomes capture the true distribution of these clades.

For each posterior tree, we searched for all subtrees with a minimum of two tips (samples) where all tips are exclusive to one of the three regions (i.e., Australia, Wallacea, or New Guinea) in each of the 10,000 posterior trees. All possible subtrees were enumerated using the clade.members.list() function from the caper package [44] for the R statistical programming language [45] and an iterative search was performed on the resulting subtree space to extract the largest GEC containing each sample. This resulted in a set of clades with ≥2 tips for each posterior tree, with all remaining single tips and geographically dispersed clades being discarded. Of the largest GECs, we retained the estimated root age only for those clades occurring in a majority (i.e., >5000) of the 10,000 posterior trees. This ensures that each GEC has majority support in the posterior phylogenetic tree space and that each sample only occurs within a single clade, making each of the final GECs (and their TMRCAs) independent with respect to their tips. Note that, for the sake of simplicity, we classified the island of Aru as Wallacean, despite it formerly being part of Sahul prior to the sundering of the continent ~9 ka [46].

To see if any regional clusters of TMRCAs existed —which might be informative about the timing of historical population movements—we used the default kernel density estimator function, density(), from the R statistical programming language to aggregate the distributions of the TMRCAs of all GECs from each of the three regions. Because GECs contained varying amounts of TMRCA estimates (which was dependent on the number posterior BEAST trees that they appeared in), all GECs within a region were rescaled to have a total probability density equal to one (i.e., $t_{i}/\sum_{i}t_{i}$, where *t_i_* is the *i*-th density estimated for the GEC). This ensured that each GEC contributed equally to the final aggregate distribution of TMRCAs computed for each region. 

### 2.6. Migration Model Inference and Testing

Bayestraits v3.0.2 [47] was used to test different models of migration history across Wallacea, New Guinea, and Australia, based on the 10,000 posterior trees and associated parameters generated by our BEAST analyses. All 656 samples were divided into three discrete ‘states’ on the basis of their sampling location, i.e., Australia (AUS; state 1, *n* = 165), New Guinea (PAP; state 2, *n* = 392), and Wallacea (WLC; state 3, *n* = 99). We tested 16 separate models in total, ranging from the saturated model with unconstrained independent migration rates between all ordered pairs of states (i.e., six parameters, one for each ordered pair) to a null model where all rates were forced to be equal (i.e., one parameter) (Appendix A). Intermediate models either constrained specific migration rates to be dependent or 0 for a subset of ordered pairs. Parameter estimation was performed using a multistate stepping-stone MCMC sampling procedure, with posterior samples being drawn every 10,000 steps from 10 million iterations, following a burn-in phase of one million iterations. The marginal likelihood, *L_i_*, for each model was obtained using the inbuilt stepping-stone sampling method with 250 stones and 10,000 iterations for each stone [48]. The marginal likelihoods were used to compare model fit using log-transformed Bayes factors (BF)—i.e., computing 2 × [*log*_10_(*L*_j_) − *log*_10_(*L*_k_)], where *j* and *k* index the models with the higher and lower marginal likelihoods values, respectively [49]. Progressively larger BF scores provide increasing evidence that the higher scoring model provides a significantly improved fit, with BF ≤ 2 providing weak to no evidence for an improved fit, BF between 2 and 5 being considered as strong evidence, and BF ≥ 5 constituting very strong evidence [49].

### 2.7. Simulating and Estimating the Timing of Migration Events

We used the R implementation of the stochastic mapping algorithm SIMMAP [50], (i.e., make.simmap(); available in the phytools package [51]) to generate 1000 realisations of the state transition process along the consensus BEAST tree for the four top performing models with predicted Wallacean ancestral nodes that were identified in BayesTraits (which had improved fits relative to all other models; see Section 3). By aggregating the timing of the state transitions across all simulations, this method provides a distribution of migration event times (for each ordered pair in the model) that is conditional on the topology and tip states (i.e., sampling region) of the consensus BEAST tree. For each model, the transition rate matrix, Q, was either set as the empirical BayesTraits estimate or was inferred internally by setting Q = “empirical” in the make.simmap() R function. Simulations using the BayesTraits rate matrix also used the estimated ancestral node state from the same model, whereas other simulations either fixed the ancestral states to be equal (setting the function parameter *pi* = “equal”) or forced the ancestral state to be Wallacean. This resulted in three separate parameterisations for each model.

Additionally, because the SIMMAP process only infers a continuous transition rate for each modelled ordered pair of states—corresponding to the migration events occurring at a fixed rate across the entire tree, which is unlikely to match the true migration history—we weighted our estimates by accounting for the total branch length in successive time intervals across the tree. Specifically, we divided the tree into successive nonoverlapping 1000 year time intervals from the tips to the root and summed the total branch lengths in each interval. For each estimated transition rate, we calculated the expected count for each interval by multiplying the total number of transitions by the proportion of the total tree length in that interval. The corrected value was then calculated as the difference between the observed and expected number of transitions in each 1000 year time interval. 

## 3. Results

### 3.1. Summary of New Mitochondrial Haplogroups from Wallacea and West Papua

Of the 342 nonredundant mitogenomes that were newly sequenced in our study, approximately one-third (*n* = 120; ~34.2%) were from the West Papua region. The majority of these West Papuan mitogenomes (104; ~86.7%) belonged to haplogroups previously observed in Papuans—i.e., macro-haplogroups Q (*n* = 69, 57.5%), P (*n* = 27, 22.5%), M73 (*n* = 3, 2.5%), and N* (*n* = 3, 2.5%)—or Aboriginal Australians—i.e., N13 (*n* = 1, <1%) (Appendix A). The remaining 16 samples (13.3%) have previously been associated with Austronesian or other Asiatic ancestry—i.e., B4a1a (*n* = 7, 5.83%), B5b1 (*n* = 3, 2.5%), E1a (*n* = 3, 2.5%), E2a (*n* = 2, 1.67%), and M7c (*n* = 1, <1%). Notably, the Austronesian/Asiatic haplogroups were predominantly concentrated amongst the Sorong population (making up 75% of all West Papuan samples and ~40% of Sorong-specific samples) located on the west coast of the Birds Head Peninsula. In contrast, Austronesian/Asiatic-associated haplotypes only made up ~6% of the samples from the Keerom population of northern West Papua and were entirely absent from the Mappi population from inland West Papua (Appendix A), which is consistent with results from Papua New Guinea where Austronesian/Asiatic ancestry is largely concentrated along the coast and almost entirely absent from highland regions [4,7].

In contrast to the West Papuan populations, haplotypes associated with Austronesian/Asiatic ancestry comprised more than 60% (*n* = 141) of the 236 unique Wallacean mitogenomes—including macro-haplogroups B (*n* = 45, 19.5%), E (*n* = 33, 14.3%), M7 (*n* = 22, 9.5%), R9 (*n* = 10, 4.3%), F (*n* = 7, 3%), D (*n* = 6, 2.6%), and M74 (*n* = 6, 2.6%), with several minor haplogroups (e.g., M20, M21, M47, M72, N21, N10, and Y) at less than 1% (Appendix A). Of the remaining 90 samples, 87 have previously been observed in individuals of Australo-Papuan ancestry in PNG or Australia, i.e., haplogroups Q (23.8%. *n* = 55), P (10.8%, *n* = 25), M73 (2.2%, *n* = 5), and R14 (1.7%, *n* = 4). The remaining three samples have a previously unreported haplotype M23′75, which shares a single mutation (A12279G) with haplogroups M23 and M75 that are found in East Asia [52] and Madagascar [53], respectively. Notably, the proportion of Australo-Papuan haplogroups observed in each population varied widely, being entirely absent from the Vayu population from central Sulawesi and comprising between 17% and 83% of total haplotypes in the remaining seven populations (Appendix A). 

### 3.2. Phylogeographic Analyses

To further discriminate the historical genetic relationships among Wallacea, New Guinea, and Australia, we used the BEAST2 software to estimate the phylogenetic structure of a combined set of 656 mitogenomes sampled from Indigenous populations across these three regions (see Section 2, Figure 1 and Appendix A). To focus our inference on the phylogeographic history specific to these three regions, we excluded all samples with mtDNA haplotypes previously associated with Austronesian or other Asiatic ancestries from our BEAST analyses. Much of the genealogical information contained in samples bearing Austronesian/Asiatic haplogroups arose prior to the arrival of these lineages in Wallacea, New Guinea, and Australia, which is not relevant for the long-term local genetic history that we focus on here. We also chose to include two haplogroups where the geographical affiliations were less clear—i.e., the M23′75 haplotype (three samples all from Rote-Ndao) that has not previously been reported and the single N10 haplotype from Sanana that also occurs at low frequencies in East Asia [52,54].

The resulting BEAST consensus tree (Figure 2 and Appendix A) exhibits the same marked population structuring between Indigenous Australian and Papuans reported in previous studies [5,6], which is characterised by distinct haplotype groups specific to each region that form deep roots within the consensus tree. The internal structuring and dating of ancestral nodes associated with these Australo-Papuan mtDNA lineages also agree with previous results, with TMRCAs for major autochthonous haplogroups dating between 40–60 ka, supporting a separation between Australian and Papuan mtDNA lineages ~50 ka ago (Figure 2 and Appendix A), close to the arrival of AMH in the region inferred from genetic data [5,6]. Strikingly, we see only one deeply rooting clade that is specific to Wallacea (i.e., M23′75 haplogroup; Figure 2). Other than this clade and a single individual with a basal P* haplotype (Figure 2), all other Wallacean samples form polyphyletic clades with Papuan samples or mixtures of Australian and Papuan samples (Figure 2). Notably, this pattern is not expected if the Wallacean mitogenomes are derived forms of the mtDNA haplotypes found amongst the initial AMH migrants in Wallacea ~50 ka ago—in which case we would expect additional basal branches similar to those observed for Australian and Papuan individuals. Thus, the lack of these deeply rooted Wallacean lineages suggests that the majority of mitogenomes in the present study resulted from one or more back-migrations of Australo-Papuan women into Wallacea subsequent to the initial peopling of the region ~50 ka ago. 

To place these results into a formal phylogeographic context, we extracted TMRCA information from all geographically exclusive clades (i.e., GECs; subtrees with ≥2 tips that come exclusively from either Wallacea, New Guinea, or Australia; see Section 2) found in the 10,000 posterior BEAST trees. Furthermore, we restricted our analyses to clades occurring in the majority (>5000) of these posterior trees, thereby ensuring that each individual sample only occurred in a single clade.

Performing this procedure on the 10,000 posterior trees produced 68 distinct GECs—35 in New Guinea, 21 in Wallacea, and 12 in Australia—with nodal support ranging from 50.9% to 99.5% (median = 88.5%; Figure 3 and Appendix A). After combining the TMRCAs from all GECs in each region, two distinct clusters occurring at ~2–3 ka and ~14–16 ka ago were clearly evident for Australia, New Guinea, and Wallacea (Figure 4A and Appendix A). Notably, both of these modes were less pronounced for Australia, which had only two GECs (~17%) with median TMRCA <10 ka ago, and three (25%) with a median TMRCA between 10 and 20 ka ago. In contrast, all GECs from Wallacea and the majority of Papuan GECs had TMRCAs that were concentrated within these two intervals (Wallacea: ~52% <10 ka ago and ~48% between 10 and 20 ka ago; New Guinea ~34% <10 ka ago and 51% between 10 and 20 ka ago; Figure 3 and Appendix A). Accordingly, the Australian GECs had a TMRCA distribution with a substantial tail of deeper dates that was not apparent in either New Guinea or Wallacea; indeed, Wallacea had no GEC with a median TMRCA that predated 20 ka ago (oldest median TMRCA ~18 ka ago), and New Guinea only had four such GECs in total (~14%; oldest median TMRCA ~26 ka ago). This resulted in Australia having an estimated 90th percentile TMRCA (~42 ka ago) that was approximately twice as old as the comparative values observed for New Guinea (~23 ka ago) and Wallacea and (~18 ka ago) (Figure 4B). Furthermore, while the estimated TMRCA for each GEC was positively correlated with the number of individual samples observed within each GEC, the TMRCA was consistently older for Australia than both New Guinea and Wallacea after accounting for the number of samples (Figure 4C). This suggests that the tendency toward older Australian TMRCAs was not caused by differences in the distribution of GEC sizes across the three regions.

### 3.3. Estimating Rates and the Timing of Historical Movements

The clustering of TMRCAs is suggestive of temporally contracted periods of mtDNA lineage movement across Australia, New Guinea, and Wallacea that occurred broadly contemporaneously across the three regions. To provide more information on these putative demic movements, we used the Bayesian software BayesTraits [47] to test the fit of 16 different models of historical population migrations across the three different regions (Appendix A). In this model, each of the three regions was defined as a discrete ‘state’ that evolved at a specific rate along the tree, with a state change on a specific branch being analogous to a movement of the corresponding mtDNA lineage from one region to another. The most complex model allowed for different migration rates between all six ordered pairs of regions (ALL in Appendix A), with other models imposing one or more constraints on movements (e.g., rates forced to be 0 or equal between pairs of regions). Model fit was evaluated against a null model that constrained all movements to a single rate (Appendix A).

Our analyses suggested that four models, including the null model, fit the phylogenetic data equally well and all offered an improved fit (Bayes factor ≥ 2) when compared with the remaining models (Appendix A). However, amongst the four best-fitting models, none had significantly improved fit to the data relative to the null model (BF < 2 for all pairwise comparisons; Appendix A), consistent with a general lack of phylogeographic structure informative about the initial colonisation of Sahul. Nonetheless, two of the four best-fitting models (model 4 and model 9; Appendix A) inferred the ancestral state to be Wallacea, suggesting that some information about the initial peopling of Sahul may have been retained in the samples. Both models allow for movements from Wallacea to Australia or New Guinea, as well as from New Guinea to Wallacea (prohibiting all other movements), and they differ only in whether the New Guinea–Wallacea migration rates were constrained to be equal or not (equal rates imposed in model 9). The only other model with a higher likelihood relative to the null (model 12; Appendix A) differed from models 4 and 9 in that it includes bidirectional movement between Australia and New Guinea, but does not allow movement between Wallacea and Australia, whereby any exchanges between these two regions must have initially traversed through New Guinea. The ancestral state was inferred as Australian for model 12, which was also the inferred ancestral state for the null model and the most commonly inferred ancestral state across all models (56%, Appendix A), likely reflecting the deeper TMRCAs found in this region.

While the BayesTraits procedure provides information on the migration rates, it does not explicitly identify when these movements were taking place. To estimate the timing of putative underlying population migrations, we used the R software implementation of the stochastic mapping method, SIMMAP [50], to provide realisations of the state transition process across the BEAST consensus tree. In this case, a state transition is analogous to a migration event between two regions, and its location on the tree is a single estimate for the timing of the underlying migration event. Accordingly, we performed 1000 simulations for each of the four best-fitting models identified by BayesTraits under different parameterisations (see Section 2), and then aggregated the estimated state transition times in successive 500 year time bins to examine if migration events clustered at specific time intervals along the tree.

Because the stochastic mapping process implemented by SIMMAP assumes that all state transitions occur at a constant rate across the tree, migration events have increasing opportunities to arise as the number of branches increases moving from the root toward the tips of the tree (i.e., approaching the present). To control for this factor, we rescaled the estimated transition rate by subtracting the expected number of transitions (which accounts for the total branch length within successive temporal intervals; see Section 2.7) from the observed count. Ultimately, we are interested in determining if there are intervals of time when more transitions were occurring than expected (i.e., after accounting for cumulative branch lengths over this interval), as this might point to specific time periods where migration was particularly concentrated.

The resulting distributions of transition times are notably coarse (Figure 5 and Appendix A), but one consistent result emerged, i.e., the migration rate of Papuan lineages into Wallacea exhibited a large peak between ~5 ka and 20 ka ago, which exceeded expectations regardless of which model or parameter set was used (Figure 5 and Appendix A). The patterning of all other migration rates is highly dependent on the underlying model, with model 12 effectively mirroring the results from the null model, and models 4 and 9 also displaying consistent patterns, but is otherwise mostly insensitive to the underlying parameterisations (Appendix A). Notably, the similarities between the null model and model 12 appeared to stem from a general lack of movements between Wallacea and Australia being inferred under the null model, despite both movements being possible (Appendix A). Accordingly, the lack of these events resulted in the null model having qualitatively similar migration patterns to those in model 12, in which movements between Wallacea and Australia were constrained to be 0. 

## 4. Discussion

Our new Wallacean and West Papuan mitogenomes provide further confirmation of the early peopling of Sahul around ~50 ka ago, but were uninformative about the potential path(s) taken by AMH en route to Sahul. Deep-rooted Wallacean clades—which could have provided the phylogenetic signals needed to infer these potential paths—were almost entirely absent from the phylogenetic history of the tested samples. Accordingly, BayesTraits and SIMMAP analyses provided weak evidence suggesting that both New Guinea and Australia were equally likely points of entry and were likely underpowered due to a lack of informative Wallacean mtDNA lineages. In contrast, our phylogeographic analyses provided unexpectedly rich insights into the migratory history of Wallacea, New Guinea, and Australia that followed the initial settlement of AMH. In particular, we observed two clusters of TMRCAs that appeared across all three regions, which closely align with historical periods of warming that followed the termination of the Last Glacial Maximum (LGM) in the Southern Hemisphere ~18 ka ago [7,55], as well as the arrival of Austronesian-speaking seafarers into Island Southeast Asia and Oceania ~3–4 ka ago [56], implying that these were key periods of demographic change and movement across the region. 

### 4.1. Post-LGM Population Expansions and Movements

A recent study of >300 genomes from modern populations across PNG [7] found that the population structure in Highland groups most likely arose within the last 10 ka—possibly associated with expansion of the Highland plant cultivator populations [57], whereas structuring between Highland and Lowland PNG populations arose earlier, between 10 and 20 ka ago. While no West Papuan populations were included in the PNG genome study, in the present study, we observed that the median TMRCAs of Papuan GECs that included West Papuan samples also predominantly fell between 10 and 20 ka ago (including GECs entirely composed of West Papuan samples; Appendix A), suggesting that the population structuring observed for PNG extends more generally across all of mainland New Guinea. Similarly, a previous analysis of a largely separate set of >100 Aboriginal mitogenomes also revealed patterns consistent with multiple migrations into the Australian interior around the end of the LGM, but found no evidence for contemporaneous events in more coastal regions [5]. These results suggest that the post-LGM warming period coincided with a period of wholesale genetic restructuring across mainland New Guinea, but had a more regional impact in Australia, despite these two regions comprising the northern and southern sections of the contiguous Sahul landmass during this period.

The alignment of the putative population expansions across New Guinea and Australia with the end of the LGM suggests that these movements were facilitated by warmer and wetter conditions that prevailed across much of post-LGM Sahul [58]. Information on the climatological conditions in Wallacea during and after the LGM is more limited, but research suggests that the end of the LGM also marked a general transition from cool and dry to wet and warm climates across this archipelago [59]. While this suggests that the observed cluster of TMRCAs around 15 ka ago for monophyletic Wallacean clades was also largely a result of climate-driven population expansions taking place throughout Wallacea, our analyses suggest that other factors may have been more important. Most notably, Wallacean mitogenomes tended to be nested amongst Papuan samples within the BEAST consensus tree (Figure 2) and Wallacean GECs lacked TMRCAs beyond ~17,700 ka ago, whereas Papuan and Australian GECs had ~29% and ~67% of their TMRCAs falling after this time, respectively (Figure 3). Furthermore, our SIMMAP analyses pointed to Papuan migration into Wallacea becoming elevated around the end of the LGM, but being absent before this point (Figure 5). These patterns suggest that the majority of Wallacean mitogenomes were derived versions of Papuan lineages, which were likely introduced into Wallacea in the period following the LGM. Notably, a recent genomic study of population genetic history of the Philippines detected the introgression of Papuan genetic lineages into southeastern Philippine populations from 15 ka [60], and our mtDNA data also support the movement of mainland Papuan Q1 haplogroup into Island Melanesia around this time (i.e., clade 12 in Figure 3 and Appendix A). These results suggest that the internal population expansions occurring in mainland New Guinea around the end of the LGM also coincided with the movement of Papuan genetic lineages into neighbouring islands that might have also included the Southeastern Philippines.

### 4.2. Redistribution of Papuan mtDNA Lineages Following Austronesian Contact

The other striking phylogenetic signal revealed in the present study was the large excess of TMRCAs occurring ~3–4 ka ago in GECs across Wallacea, New Guinea, and Australia (albeit a more moderate signal in the latter), which coincides with the arrival of Austronesian seafarers into these regions [56]. The dispersal of Austronesian-speaking populations saw the introduction of Asiatic mtDNA haplogroups across islands in Southeast Asia, New Guinea, and beyond [22,23,24], which were also observed amongst the new Wallacean and West Papuan mitogenomes reported here. The distribution of Austronesian-affiliated haplotypes in our samples closely align with prior evidence, which support a substantial Austronesian genetic footprint across modern Indonesia [26], with more limited genetic impacts in New Guinea that were largely confined to coastal regions [7,27].

Intriguingly, no convincing genetic evidence currently exists for Austronesian introgression into Indigenous Australian groups, including the mtDNA samples used in the current study. However, we did observe two Australian GECs with median TMRCAs around 3.5 ka ago (clades 11 and 12 in Figure 3 and Appendix A); both clades contained pairs of individuals reported to have Torres Strait Islander ancestry [61] who carry haplotypes affiliated with mainland Papuan lineages (i.e., P2a and P1e). Because Wallacean GECs with TMRCAs < 5 ka also bore derived forms of mainland Papuan haplogroups, our results suggest that the arrival of Austronesian seafarers facilitated the redistribution of mainland Papuan mtDNA lineages more widely across the local region. This has been suggested previously for Timorese populations [62], but our results indicate that this was a more general phenomenon that minimally affected multiple islands across Wallacea and the Torres Strait. Previous research suggests that the migration of Austronesian-speaking peoples likely comprised periods of cultural and demic transfer that resulted in the incorporation of local genetic diversity into an expanding Austronesian gene pool [63]. Our results imply that such an integration process also likely facilitated the movement of historically mainland Papuan mitochondrial lineages throughout neighbouring islands in Wallacea and the Torres Strait; however, it has left no detectable genetic impact on mainland Aboriginal Australian populations surveyed to date.

### 4.3. Comparison with Wallacean Archaeological and Linguistic Records

While Wallacean archaeological surveys and excavations have largely been limited to Sulawesi and a handful of islands across the Moluccas and eastern Lesser Sundas, the growing body of evidence suggests that the broader archipelago was impacted by a series of major technological and demographic changes concentrated around the end of the LGM [64]. Several distinct AMH technologies make their initial appearance in Wallacea in the post-LGM period, including single-piece shellfish hooks, edge-ground adzes/axes made on large *Tridacna* shells, and double-holed *Nautilus* beads [64,65,66]. Wallacean exchange networks also become apparent around this time, as exemplified by the inter-island spread of these new technologies and the appearance of exotic obsidian in the Lesser Sundas around 15–13 ka ago from an as yet unidentified source [67,68]. The end of the LGM also marked a notable intensification of AMH occupation in Wallacea [19,69,70] with increased usage observed at previously occupied sites and multiple Wallacean islands recording either an increase in the number of occupied sites (e.g., Rote and Alor) or the first site occupation on the island (e.g., Kisar, Morotai, and Obi) [71]. These patterns suggest that increased population mobility and growth may have been a common feature of post-LGM Wallacea. Current interpretations suggest that these changes were spurred by the availability of marine resources along new coastal margins and, crucially, the development of improved watercraft technologies (possibly dug-out canoes) fashioned using shell adzes that enabled more reliable and regular inter-island maritime travel [64,72]. 

Another notable change in the Wallacean technological repertoire occurred ~3 ka ago, with of a new type of adze made on *Cassis* shell first appearing in deposits in the Lesser Sundas (Flores) and the Moluccas (Gebe) at this time [73,74,75,76]. Notably, worked *Cassis* adzes are also common in contemporaneous sites in Island Melanesia and the broader Pacific, emphasising a connection to Wallacea from the east during this period [76]. Similarly, *Trochus* shell armbands, which are common in Lapita assemblages in Papua New Guinea [77,78,79] first appeared in Wallacea between 2 and 2.7 ka ago [75,76,80,81], and the New Guinea cuscus (*Phalanger* species) was likely first introduced into Wallacea around 3.3 ka ago [82,83], further supporting the movement of people and culture from Melanesia to Wallacea coincident with the arrival of Austronesian-speaking seafarers in the region. 

Further evidence for the historical influx of Papuan migrants in Wallacea comes from contemporary populations in the Moluccas and eastern Lesser Sunda Islands that speak languages associated with the Trans New Guinea (TNG) language phylum; however, there is some debate regarding when these languages first arrived in Wallacea. For instance, it has been suggested that the TNG language Banuq now spoken in central Timor first arrived prior to Austronesian contact [84], while others argue that another TNG language, Fataluku, now spoken in eastern Timor-Leste replaced an Austronesian language, Lovaia, that was previously spoken in this area [66,85,86]. Although currently undated, shared art motifs found in the MacCluer Gulf region of West Papua and Fataluku speaking parts of Timor-Leste also attest to direct cultural transmission between these regions [87]. Given the coexistence of Austronesian and TNG languages in the MacCluer Gulf, it remains possible that both language groups were first introduced into Timor by Papuan migrants from this region ~3 ka ago, perhaps translocating the cuscus in the process.

Taken together, the archaeological and linguistic records provide further support for the movement of Melanesian lineages into Wallacea coincident with the arrival of Austronesian seafarers in the region between 3 and 4 ka ago, as well as indicating major changes across Wallacea following the LGM, including the appearance of new technological items and marked demographic transitions. While it is plausible that these demographic patterns reflect shifts in local population dynamics, the results of our mtDNA analyses suggest that they may also record the arrival of migrants from mainland New Guinea.

## 5. Conclusions

The marked phylogeographic patterns observed in our study support the arrival of AMH in Sahul ~50 ka ago [4,5,6,61] and provide further evidence for a large-scale population expansion in New Guinea around the end of the LGM that generated modern mainland Papuan genetic structure [7]. Unexpectedly, the inclusion of new genetic information from previously unsampled populations in Wallacea and West Papua did not help to elucidate details of the island routes followed by AMH en route to Sahul, but it did reveal strong evidence that the post-LGM population expansion in New Guinea also resulted in the dispersal of Papuan mtDNA lineages into the neighbouring Wallacean archipelago and may have even been the cause of the recently reported Papuan ancestry occurring in southern Philippine populations [60]. Additionally, our results build upon previous findings showing that the arrival of Austronesian-speaking seafarers facilitated an additional dispersal of mainland Papuan mtDNA haplogroups across multiple Wallacean islands, as well as into the Torres Strait Islands, although notably not amongst the relatively small number of mainland Aboriginal Australian groups currently represented in genetic research.

Importantly, not all TMRCAs observed for Wallacean clades that cluster around the post-LGM period may have been the result of migrations during this period; it is plausible that some were Austronesian-era movements that share a common ancestor in New Guinea around the putative post-LGM Papuan expansion. Nonetheless, regardless of when Papuan mtDNA lineages arrived in Wallacea, a striking impact of the combined post-LGM and Austronesian-era demic movements has been the effective swamping or replacement of autochthonous mtDNA lineages across Wallacea with mainland Papuan- and Asiatic-affiliated haplotypes. Indeed, only four of the 351 Wallacean mitogenomes (~1%) generated for this study were found to root basally in the BEAST consensus tree (Figure 2), which currently comprise the sole evidence for autochthonous Wallacean mtDNA lineages. Crucially, the lack of autochthonous Wallacean lineages is also the most likely explanation for the absence of evidence for the possible migratory routes followed by AMH into Sahul; we simply lack sufficient autochthonous Wallacean mitogenomes to recover the phylogeographic patterns that are informative about this process.

The current absence of archaeological data across much of Wallacea means that it is unclear whether the preponderance of Papuan mtDNA lineages in modern Wallacean populations represents the recolonisation of uninhabited islands, admixture with relatively small local populations, competitive replacement, or some mixture of these processes. Achieving a better understanding of these historical population processes is also essential for clarifying several intriguing results from previous research in the region. In particular, a recurrent finding from genomic studies of modern Indonesian populations has been the clinal distribution of Papuan ancestry that increases from west to east across the archipelago, complemented by a counter-gradient of Asiatic-derived ancestry, a large component of which is inferred to be Austronesian [24,26,27]. This gradient of Papuan ancestry has generally been interpreted as the genetic substrate arising from the original settlement of the region by AMH [27]; however, the present study suggests that much of this ancestry and the resulting clinal pattern could instead be mostly the result of Papuan backflow arising after the LGM. Additionally, modern human genomes from ISEA, New Guinea, and Australia contain traces of multiple separate historical admixture events with now-extinct archaic hominin species [6,88,89,90], and Wallacea has been identified as a plausible location of some of these events [91]. Unravelling the timing and location of these admixture events and the more general population history of Wallacea, Sahul, and the wider region awaits more powerful genomic analyses that are appropriately integrated within existing archaeological, linguistic, and climatological datasets.

## Figures and Tables

**Figure 1 genes-12-00965-f001:**
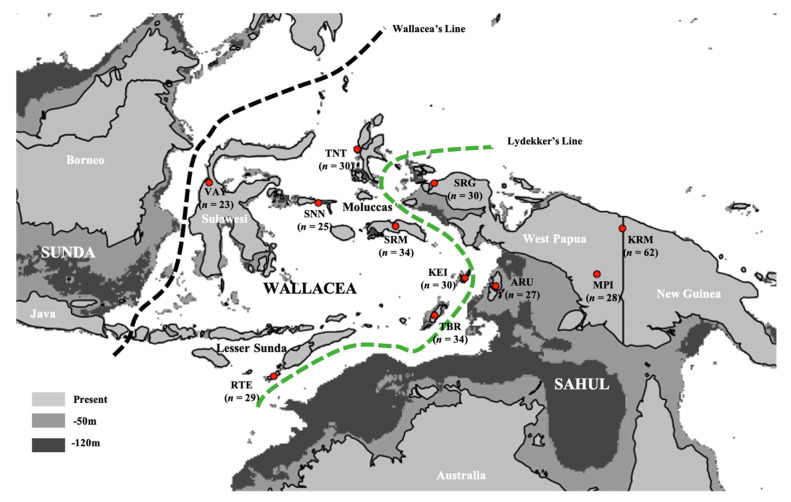
Historical land coverage across study regions and sampling locations of new mitogenomes. Exposed land areas for the Sunda and Sahul continental shelves and the interceding Wallacean islands are shown for the current period (light grey) and inferred for lower sea stands occurring ~65 ka (−50 m; medium grey) and ~20 ka (−120 m; dark grey) before the current period [13]. The locations of major biogeographic boundaries separating Wallacea from the Sunda and Sahul shelves are shown as black (Wallace’s Line) and green dashed lines (Lydekker’s Line), respectively. Sampling locations (red dots) are indicated by their abbreviated population IDs (see Appendix A), with sample size indicated directly beneath.

**Figure 2 genes-12-00965-f002:**
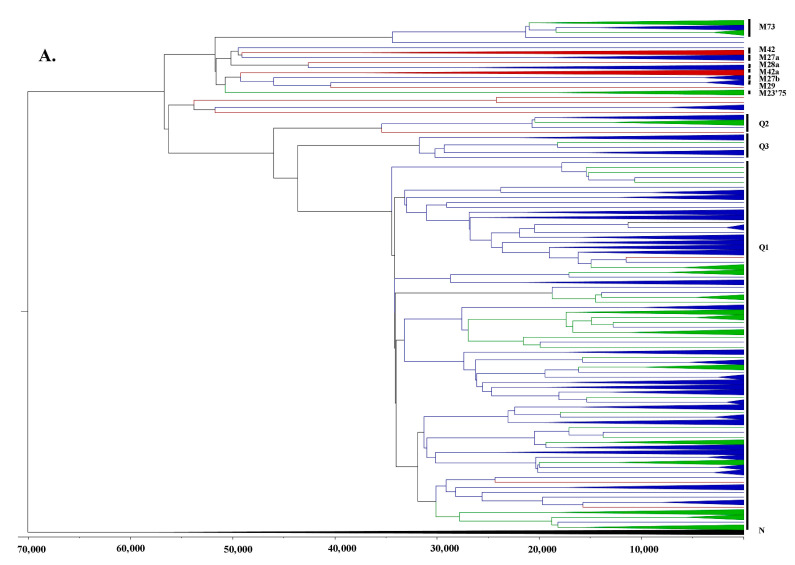
BEAST consensus tree for 656 mitogenomes from Wallacea, New Guinea, and Australia. The tree is based on putative Australo-Papuan-affiliated haplogroups, split into the two main non-African macrohaplogroups M (**A**) and N (**B**). Samples from Wallacea (WLC) are coloured green, with those from Papua (PAP; i.e., New Guinea and Island Melanesia) and Australia (AUS) being coloured blue and red, respectively. All monophyletic clades are collapsed.

**Figure 3 genes-12-00965-f003:**
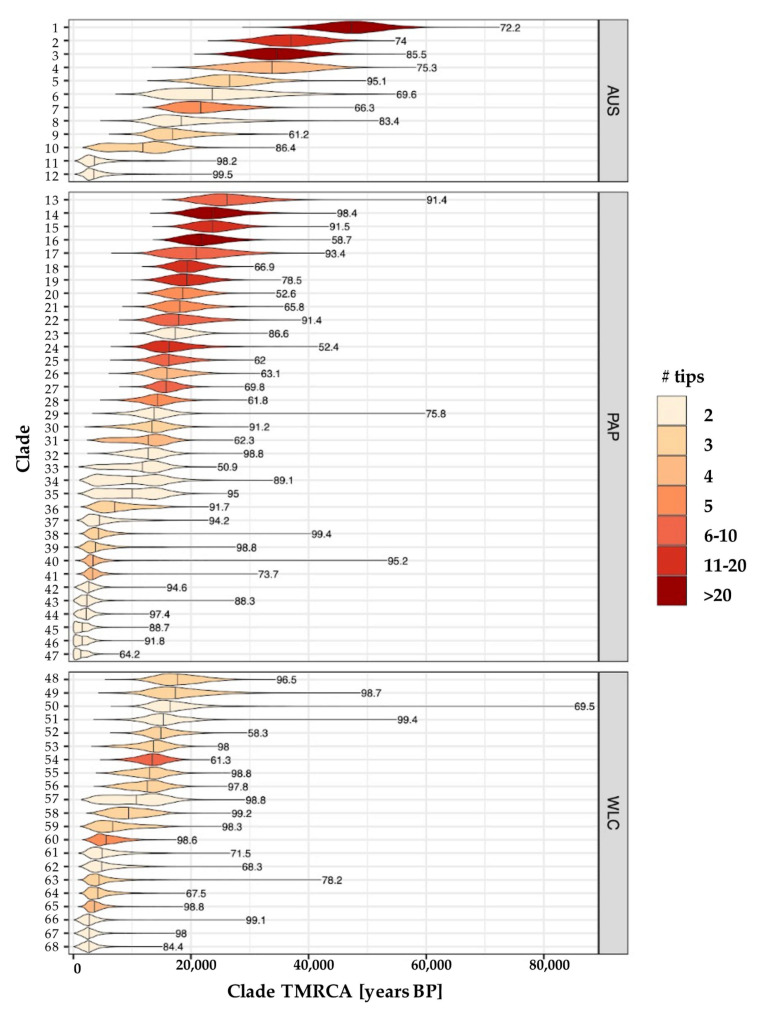
Distribution of TMRCAs for all monophyletic clades. Distributions are shown for each of the 68 monophyletic clades with >50% support (support shown in text to the right of each distribution). Clades are organised by region—AUS = Australia; PAP = Papua (i.e., New Guinea and Island Melanesia); WLC = Wallacea—and are coloured according to the number of tips (samples) that they contain. Vertical lines of each distribution indicate median estimated TMRCA. Additional information for each clade is provided in Appendix A and indexed according to integer labels on the *y*-axis in the current plot.

**Figure 4 genes-12-00965-f004:**
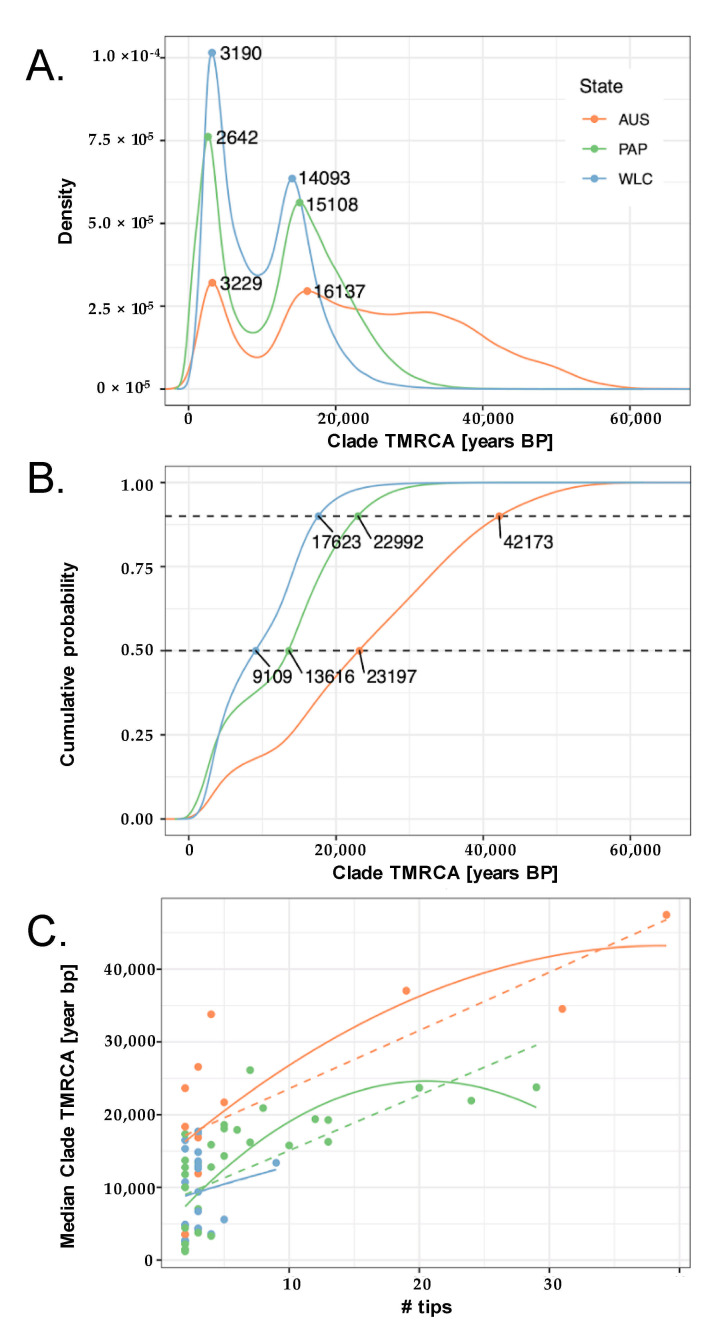
Properties of aggregated TMRCAs across Wallacea, New Guinea, and Australia. Probability distributions (PDF; (**A**)) and cumulative distributions (CDF; (**B**)) for the TMRCAs aggregated across all monophyletic clades in each region (see key in (**A**)). TMRCAs for the 50th and 90th percentiles are indicated by the lower and upper dashed lines, respectively, in (**B**). TMRCAs were deeper on average for Australia, even after controlling for the number of tips (samples) in the clade (**C**). Linear and quadratic regression fits are shown as dashed and solid lines, respectively.

**Figure 5 genes-12-00965-f005:**
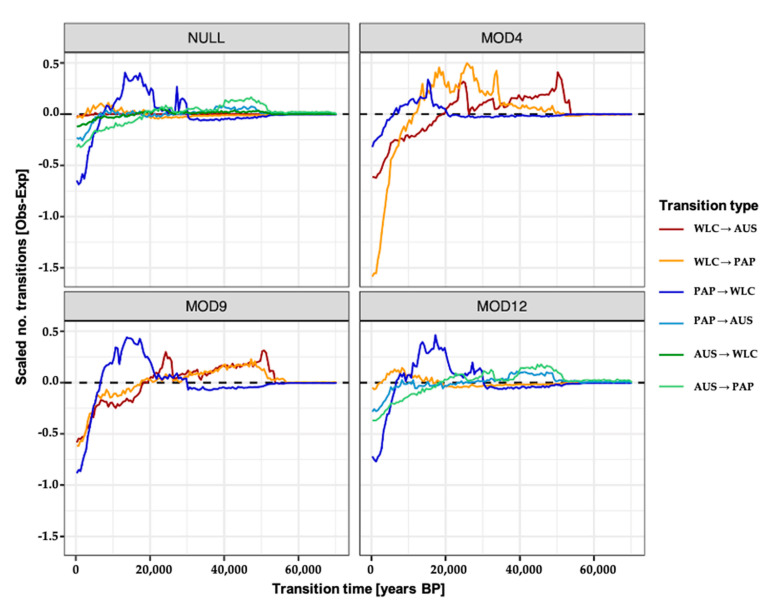
Estimated migration rates across Wallacea, New Guinea, and Australia. Distribution of the mean migration rates across each region across time for the four best-performing BayesTraits models. For each model, migration rates were estimated from 1000 SIMMAP stochastic mappings, with each migration event comprising a single simulated state transition occurring on a branch of the consensus BEAST tree. Transition rate matrices and ancestral node states were derived from BayesTraits (see Appendix A, for other parameterisations). Migration rates are scaled to account for the expected number of migration events (see Section 2).

## Data Availability

The new 351 complete mitochondrial DNA sequences have been deposited in Genbank (https://www.ncbi.nlm.nih.gov/genbank) under accession numbers XXXXXXXX-XXXXXXXX.

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
