# Peer review of "Mitogenomes Reveal Two Major Influxes of Papuan Ancestry across Wallacea Following the Last Glacial Maximum and Austronesian Contact"

_genes, 2021, doi:10.3390/genes12070965_

Round 1

Reviewer 1 Report

The authors have done an excellent job presenting information on the ancestry of an area that is generally considered to be understudied. The article is clear and well written and the methods are sound. The conclusions too are warranted based on the data. Unless another reviewer has an issue with the Bayesian analyses that I do not spot (I am aware of the method, but have not used it in my research).  

Author Response

We thank reviewer 1 for their kind comments.

Reviewer 2 Report

The authors identified nearly 300 new mitogenome sequences from Australia, New Guinea and Wallacea. Joined with an already existing sample of previously published sequences, they provide analyses on a consensus tree to infer the demographic history and potential migrations. Because population events in historical periods have been subject of previous studies, they focused on the identification of events between the Last Glacial Maximum (18 ka) and younger episodes. In their study the authors successfully identified and characterized expansion events in Papua New Guinea leading into Wallacea, possibly as far as the southern Philippines.

Although demographic links between the three regions under study are not unexpected, the detailed resolution and the direction of the migration events identified provide exciting insight into the complexity of those processes nearly 20,000 years ago.

I do not have practical experience as a geneticist. I am thus unable to comment on the technical parts of the paper. The methods, algorithms and procedures which are used for the analysis of their datasets are however, state of the art. The protocols are introduced in a convincing way and well explained.

The arguments in the discussion are stated clearly and the data supports the conclusions drawn.

Figures and Tables are necessary and carefully designed.

Author Response

We thank reviewer 2 for their kind comments.

Reviewer 3 Report

Major comments

1, Nowadays most of human DMA evolution studies are conducted using nuclear DNA data, not mtDNA data. In this sense, this paper, although well written, may not be significantly contribute to the human evolution in the region authors focused.

2. I do not understand why authors removed all mitogenomes affiliated with either Austronesian or mainland Asiatic ancestry (lines 166-167 of page 4). These sequence data should have been included in analysis.

Minor comments

  1. In Abstract, sample size used in this study are written somewhat unbiguously; “>650 whole mitogenomes” (line 39 of page 1) and “including ~200 new samples” (line 40 of page 1).
  2. In Introduction, authors should include some restrictions when mtDNA data alone were analyzed, compared to using nuclear DNA data.
  3. In Figure 3, clade number (1-68) is shown, but its meaning is not clear .
  4. I am not sure about usefulness of “geographically-exclusive clades (GECs)” extensively used in this manuscript.
  5. In Discussion, authors compared their results based on mtDNA data analyses and those for archaeological and linguistic data. However, authors never discussed nuclear DNA data. This discussion should be included.

Round 2

Reviewer 3 Report

I am disappointed with author reply.